# Structural basis of inhibition of human Na$_V$1.8 by the tarantula venom peptide Protoxin-I

Bryan Neumann ⓘ , Stephen McCarthy & Shane Gonen ⓘ ✉

Voltage-gated sodium channels (Na$_V$s) selectively permit diffusion of sodium ions across the cell membrane and, in excitable cells, are responsible for propagating action potentials. One of the nine human Na$_V$ isoforms, Na$_V$1.8, is a promising target for analgesics, and selective inhibitors are of interest as therapeutics. One such inhibitor, the gating-modifier peptide Protoxin-I derived from tarantula venom, blocks channel opening by shifting the activation voltage threshold to more depolarized potentials, but the structural basis for this inhibition has not previously been determined. Using monolayer graphene grids, we report the cryogenic electron microscopy structures of full-length human apo-Na$_V$1.8 and the Protoxin-I-bound complex at 3.1 Å and 2.8 Å resolution, respectively. The apo structure shows an unexpected movement of the Domain I S4-S5 helix, and VSD$_I$ was unresolvable. We find that Protoxin-I binds to and displaces the VSD$_{II}$ S3-S4 linker, hindering translocation of the S4$_{II}$ helix during activation.

Voltage-gated sodium channels (Na$_V$s) are integral membrane proteins responsible for the selective permeation of sodium ions into cells in response to membrane depolarization. The small differences in sequence that characterize the nine human Na$_V$ subtypes (hNa$_V$1.1-1.9, Supplementary Fig. 1) nonetheless give rise to distinct electrophysiological properties that, together with varying expression levels in different tissues, give each hNa$_V$ isoform particular physiological roles.

Na$_V$1.8, one of the three tetrodotoxin-resistant Na$_V$s, is distinguished from other isoforms by the relatively depolarized voltage-dependency of activation and inactivation, slower inactivation kinetics, and an increased persistent current[1-3]; These attributes make Na$_V$1.8 principally responsible for inward currents during the rising phase of the action potential[4,5], and contribute to hyperexcitability and repetitive firing in the dorsal root ganglion (DRG) neurons where it is primarily localized[6,7]. Uniquely, it maintains its gating properties at cold temperatures[8].

Multiple studies have linked Na$_V$1.8 to nociception and chronic pain. Gain-of-function mutations in Na$_V$1.8 causing increased excitability of DRG neurons have been identified in patients with peripheral neuropathy[9,10], while a loss-of-function Na$_V$1.8 mutation has been linked to reduced pain sensation[11,12]. Na$_V$1.8 has also been linked to inflammatory pain[13]. Studies of Grasshopper mice (*Onychomys torridus*) showed that their insensitivity to pain induced by the venom of the Arizona bark scorpion (*Centruroides exilicauda*) derives from mutations in their Na$_V$1.8 channels[14]; injection of the venom reduced the *O. torridus* pain response to the formalin test, demonstrating that inhibition of Na$_V$1.8 is a viable analgesic strategy[15].

Inhibitors of Na$_V$1.8 are therefore of interest as pain treatments, and peptides derived from animal venom are renowned modulators of Na$_V$ activity. Unlike small-molecule inhibitors, which typically bind in the highly conserved pore domain (PD), peptide inhibitors frequently bind to the less-conserved extracellular regions above the voltage-sensing domains (VSDs) which provide greater scope for isoform selectivity in drug development. Na$_V$1.8-selective peptides include scorpion venom peptide BmK I[16], the μ-conotoxins MrVIA/MrVIB[17,18] and TsIIIA[19], and the tarantula venom peptide Protoxin-I (ProTx-I)[20].

ProTx-I was isolated from the venom of the Peruvian green velvet tarantula (*Thrixopelma pruriens*) and is a gating-modifier peptide that shifts the voltage-dependence of activation to more depolarized potentials[20]. It shows slight selectivity for rat Na$_V$1.8 (IC$_{50}$ = 27 nM)[21] over other human Na$_V$ isoforms (typically IC$_{50}$ = 60–130 nM)[22], as well

Department of Molecular Biology and Biochemistry, University of California Irvine, Irvine, CA, USA. ✉e-mail: gonens@uci.edu

as activity against T-type calcium channels[23] and the TRPA1 channel[24]. ProTx-I is disulfide-rich and shares the inhibitor cystine knot (ICK) framework that is common among gating-modifier tarantula venom peptides[25]. Mutagenesis studies using hNa$_V$1.7/K$_V$2.1 chimeras localized the binding site of ProTx-I to Na$_V$1.7 on the extracellular loops of VSD$_{II}$ and VSD$_{IV}$[26] but the exact binding mechanism remained undetermined.

Structural characterization of venom peptides in complex with Na$_V$s is essential for understanding their pharmacological profiles and for realizing their potential as tool compounds and drugs. However, poor Na$_V$ expression yields and low local resolution for bound peptides have made these structures difficult to obtain. Full-length Na$_V$-peptide complexes determined to date are limited to Na$_V$1.2 in complex with the pore-blocking μ-conotoxin KIIIA[27], Na$_V$1.5 bound to the scorpion venom peptide LqhIII[28], and the American cockroach channel Na$_V$PaS bound to the spider venom peptide Dc1a[29]. Attempts to characterize complexes of human Na$_V$1.7 with the spider venom peptides Protoxin-II and Huwentoxin-IV produced high-resolution reconstructions of the channel but could not sufficiently resolve the peptide for modeling[30]. Chimeric channels consisting of bacterial or invertebrate Na$_V$ scaffolds onto which human Na$_V$ domains have been grafted have also been developed to address this problem. A Na$_V$Ab/Na$_V$1.7-VSD$_{II}$ chimera was used to determine the binding mechanism of Protoxin-II and Huwentoxin-IV[31], which has additionally been characterized in complex with a NaChBac/Na$_V$1.7-VSD$_{II}$ chimera[32], while a Na$_V$1.7-VSD$_{IV}$ chimera based on the Na$_V$PaS scaffold aided in characterization of the scorpion venom peptide AaH2 as well as small molecule inhibitors[33].

Mutagenesis screening and chimeric constructs can be effective in identifying channel variants with higher expression yield, but mutations can affect channel gating properties and chimeric constructs lack functional domains[34]. Experimental structures with full-length human channels are therefore preferred for rational structure-based drug design in order to minimize off-target interactions; maximizing yield and particle grid density through mammalian cell expression, biochemical and cryogenic electron microscopy (cryoEM) developments is therefore an attractive strategy.

Here, we used mammalian HEK293 cells to express full-length human Na$_V$1.8 and determined the structure, with and without ProTx-I bound, by single-particle cryoEM. Optimization of the expression and purification, and the use of monolayer graphene grids, allowed the maximum number of useable particles for data collection from as low as 1.5 L of cell culture. The final reconstructions were determined at an overall resolution of 3.1 Å for apo-hNa$_V$1.8 and 2.8 Å for the hNa$_V$1.8-ProTx-I complex. Separate classifications revealed large movements of the S4-S5 linker leading to VSD$_I$, and consequently this VSD was unresolvable. The resolution of the map in the ProTx-I region was sufficient for tracing of the peptide backbone and determination of its mechanism of binding. We anticipate that the developed protocols will be beneficial in the solution of future peptide-Na$_V$ complexes by cryoEM, and that these results will assist in the design of drugs targeting hNa$_V$1.8.

## Results
### Apo-hNa$_V$1.8
Human Na$_V$1.8 α-subunit was co-expressed in HEK293 cells together with the β4-subunit and purified as described in Methods. Na$_V$ α-subunits are frequently co-expressed with β-subunits to stabilize the protein, increase expression levels, and maintain a more native environment. Evidence suggests that hNa$_V$1.8 is capable of interacting with all four β-subunits, including β4, which affects hNa$_V$1.8 activation and inactivation thresholds[35]. However, despite co-expression, only the α-subunit was observed after purification (Supplementary Fig. 2) and in the final map; loss of co-expressed β-subunits has been observed for other Na$_V$s[36–38] and may reflect weak binding affinity between the proteins (see Discussion). To maximize the final hNa$_V$1.8 yield, several parameters were screened (see Methods) and the overall time between

solubilization and purification was minimized. Prior to cryoEM, particle purity and homogeneity were confirmed by negative stain (Supplementary Fig. 2d).

Screening freezing conditions using conventional holey carbon grids showed a preference for particles to accumulate over the carbon film, and low particle density in the holes (Supplementary Fig. 3a). Due to low yields, we attempted to increase the scale of the cell culture, but this introduced problems with solubilization and did not sufficiently increase the final usable protein concentration. The initial grid screening included a range of grid types, including one with a support film of monolayer graphene which showed improved particle distribution across the grid holes (Supplementary Fig. 3a). We therefore pursued the use of support film grids, including monolayer graphene, as an alternative to mutagenesis or large increases in the scale of cell culture. Using ultrathin (2–3 nm) carbon grids under similar conditions failed to produce a cryoEM dataset that could reach high resolution which we hypothesize was due to contrast loss resulting from particle packing. Graphene oxide grids indicated acceptable Na$_V$ particle density and contrast but were more susceptible to breakage from glow discharging (Supplementary Fig. 3b). All maps in this manuscript resulted from the use of monolayer graphene grids (0.4 nm thickness) which showed good particle distribution and contrast (Supplementary Fig. 3c); this allowed us to reconstruct the structures using just ~0.15 mg/mL of purified protein from as low as ~20 g wet cell pellet or 1.5 L of cell culture, significantly lower than typical Na$_V$ preparations[37,39–41].

This approach allowed us to reconstruct apo-hNa$_V$1.8 at an overall resolution of 3.1 Å (Fig. 1; Supplementary Figs. 4 and 5; Supplementary Table 1). Typical of other apo-Na$_V$ structures, the apo model shows features characteristic of an inactivated channel, with gating charge residues on all visible VSDs showing 'up' conformations (Supplementary Fig. 5) and the IFM fast inactivation motif on the VSD$_{III}$-VSD$_{IV}$ linker buried in its binding site between S6$_{IV}$, S5$_{IV}$ and the VSD$_{III}$ S4-S5 linker (Fig. 1a, c). Similar to a previous study[36], the quality of the map allowed modeling of N-linked glycosylation at several positions (Asn residues 312, 819, 1312, 1328, and 1336) on the extracellular loops, as well as possible cholesterol, lipids, and detergent molecules bound to the transmembrane region; similar to other cryoEM structures of Na$_V$ channels we observe density for a bound molecule in the intracellular pore region, which we putatively assign as cholesterol (Fig. 1c, e)[27,34,36,39,40].

Density corresponding to VSD$_I$ is almost entirely absent in the 2D and 3D classifications, and in the final reconstruction (Fig. 1b, c), even as portions of the N-terminal domain (NTD) can be observed at lower map thresholds; this result is consistent with a prior report[36]. Attempts to improve the resolution in this region through 3D classifications steps, masking, and local refinements did not improve interpretability of VSD$_I$ but did reveal separate classes (denoted Class I and Class II, Supplementary Fig. 4) showing a distinct repositioning of the VSD$_I$ S4-S5 linker and a smaller movement of the lower portion of the VSD$_I$ S6 helix (Fig. 2, Supplementary Movie 1); The final apo map and structure was calculated from all particles making up Class I and Class II. In Class I the VSD$_I$ S4-S5 linker is positioned closer to the pore domain even as S6$_I$ moves outwards (Fig. 2c, cyan arrows), while in Class II the VSD$_I$ S4-S5 linker swings outwards (Fig. 2c, purple arrows); the S6$_I$ helix moves contrarily and tucks closer into the pore. In all our reconstructions, the VSD$_I$ S4-S5 linker is positioned significantly outward (by up to 17 Å) compared to the prior apo-Na$_V$1.8 structure (PDB 7WFW)[36]. These movements necessarily affect the positioning, and likely contribute to the unresolvability, of VSD$_I$; this is supported by 3D variability analysis, where, at low thresholds, density for the NTD is observed in slightly different positions (Supplementary Movie 2). Distinct positions for NTD and VSD$_I$ have also been observed in Na$_V$1.7-M11, an engineered variant of hNa$_V$1.7 containing 11 mutations that collectively induce a large depolarizing shift in the activation voltage[42], underlining the connection between VSD$_I$ lability and activation thresholds.

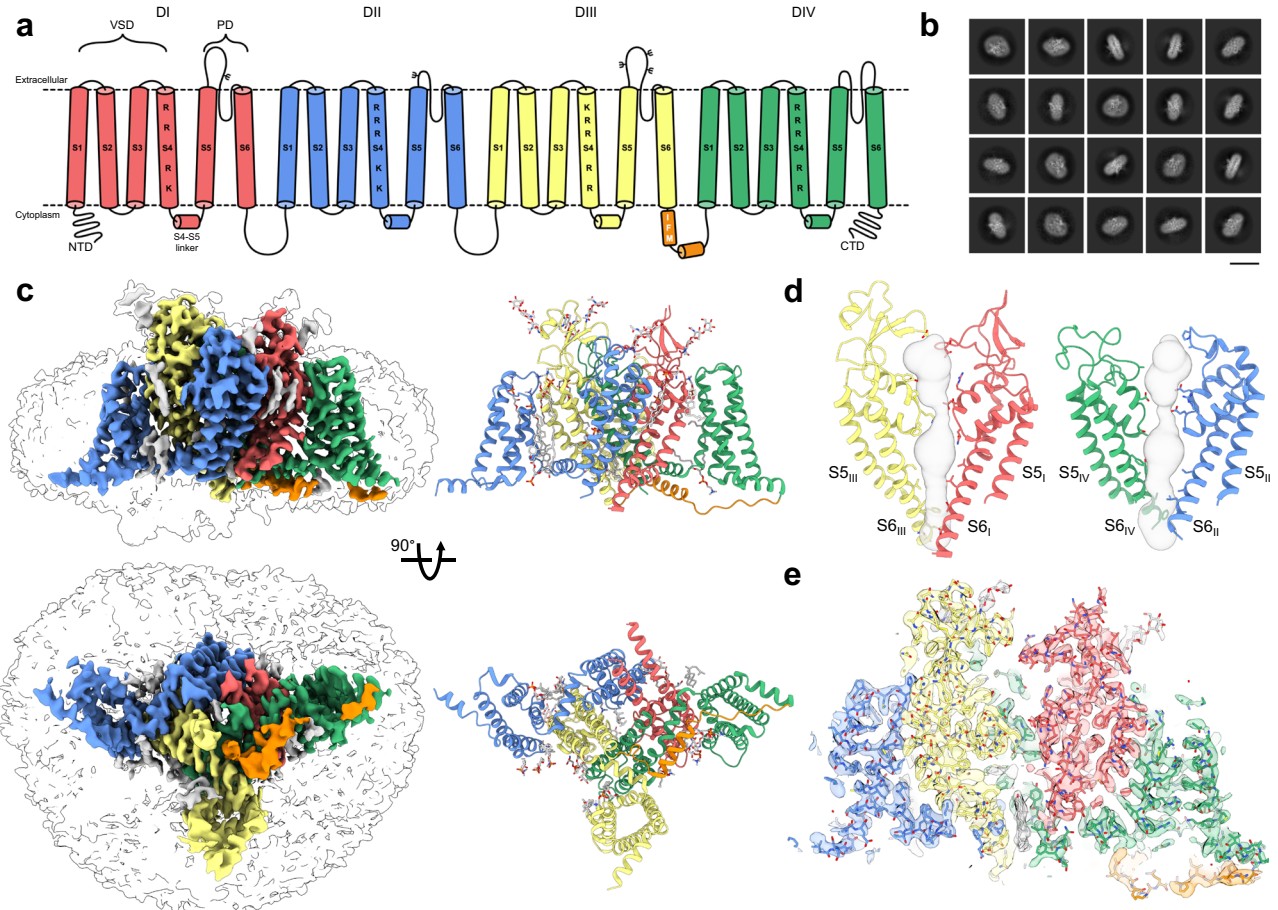

**Fig. 1 | Overall architecture and reconstruction of apo-hNa$_V$1.8. a** Topology of hNa$_V$1.8 colored by domain: DI (red), DII (blue), DIII (yellow), IFM (orange) and DIV (green); VSD = voltage-sensing domain, PD = pore domain. The basic residues responsible for voltage-sensing in S4 of hNa$_V$1.8 are labeled. hNaV1.8 glycosylation sites are labeled with the letter psi. **b** Example 2D class averages for apo-hNa$_V$1.8; scale bar = 15 nm. **c** (left) Side and intracellular views of the final apo-hNa$_V$1.8 map (colored according to the scheme in **a**) with transparent lower map threshold to indicate micelle and emerging NTD, and (right) the resulting final refined model. Glycosylation and small molecule ligands, including cholesterol in the pore are shown in gray. **d** Two views of the apo-hNa$_V$1.8 pore domain showing the ion permeation path in gray. **e** Model-to-map fit for a central cross-section of apo-hNa$_V$1.8.

Analysis of the ion permeation path shows the point of greatest restriction around the selectivity filter in the upper pore and is similar in all structures (Fig. 2a). The pore diameter through the intracellular gate is ~3 Å, consistent with the conformation of the S6 helices observed in hNa$_V$1.7[34] and which allows space for the bound cholesterol molecule in our maps.

## ProTx-I-bound complex

The hNa$_V$1.8-ProTx-I complex was prepared by incubating ProTx-I solution with purified apo-hNa$_V$1.8 prior to freezing with support film grids. We observed low particle contrast during initial screening, likely due to excess unbound ProTx-I (Supplementary Fig. 3b). We therefore introduced ProTx-I to hNa$_V$1.8 prior to the final concentration step using a high molecular weight cutoff filter to remove excess unbound peptide (as described in Methods). Processing of the data produced a map with an overall resolution of 2.8 Å from 267,708 particles (Fig. 3; Supplementary Figs. 6 and 7). Outside the ProTx-I binding region, the refined hNa$_V$1.8 structure in the complex is very similar to apo-hNa$_V$1.8, with some slight rigid-body shifts in the VSDs and small movements in the extracellular loops. As with our apo-hNa$_V$1.8 map, we observe density for multiple glycans in the map for the hNa$_V$1.8-ProTx-I complex. The higher resolution of this map, likely enabled by the stabilizing effect of ProTx-I, allowed an additional extracellular loop (ECL$_I$, D280-P295) to be traced in the hNa$_V$1.8-ProTx-I map that was not possible for apo-hNa$_V$1.8 (Supplementary Fig. 7). This loop has two further N-linked

glycosylation sites (residues N284 and N288), but the glycans could not be resolved. The VSD$_I$ S4-S5 linker is again swung outward and consequently VSD$_I$ is not resolvable, despite additional processing, a larger dataset, and the higher overall resolution of the reconstruction.

Density for ProTx-I is clearly visible in the refined map; consistent with its electrophysiological effects as a gating modifier and previous structure-activity relationship studies[26], ProTx-I binds to the S3-S4 linker on VSD$_{II}$ (Fig. 3b, d, e). The local resolution allowed the principal backbone of the peptide to be traced, which, with the assistance of the discernible β-loop near the peptide C-terminus, was sufficient to model ProTx-I into the map using an available NMR model (see Methods). The resolution of the ProTx-I portion of the map is highest immediately abutting the channel (likely due to stabilizing interactions) and is attenuated in the more distant regions (Supplementary Fig. 6c); this reduction in density in the more peripheral regions is a common feature of cryoEM studies of peptide-Na$_V$ complexes[30,33]. Evidence from mutagenesis experiments demonstrates that ProTx-I can also bind to hNa$_V$1.7 VSD$_{IV}$[26], although electrophysiological recordings have not demonstrated that ProTx-I has any effect on channel inactivation thresholds that are typically governed by VSD$_{IV}$. Despite high local resolution, we do not observe any density for ProTx-I above VSD$_{IV}$ in the hNa$_V$1.8-ProTx-I structure, noting that the S1-S2 and S3-S4 linker regions on VSD$_{IV}$ are poorly conserved between hNa$_V$1.7 and hNa$_V$1.8 (Supplementary Fig. 1).

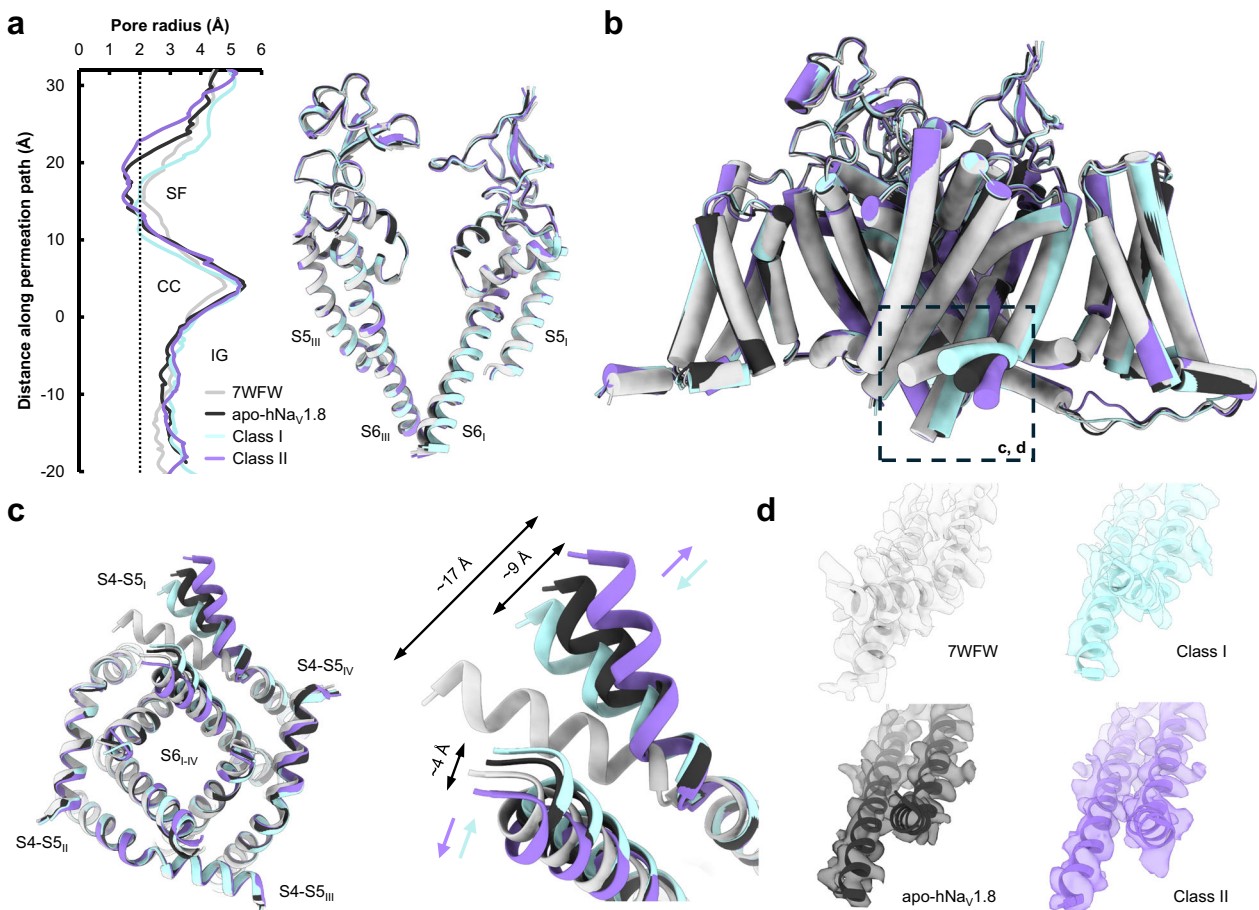

**Fig. 2 | Structural comparisons of apo-hNa$_V$1.8, highlighting dynamics of VSD$_I$ S4-S5 linker and S6 helices. a** (left) Pore radius for the overall apo-hNa$_V$1.8 model (black) together with Class I (cyan), Class II (purple), and 7WFW (light gray)[36], and (right) aligned pore domains showing minimal backbone movements. The selectivity filter (SF), central cavity (CC) and intracellular gate (IG) are indicated. **b** Comparison of the four apo-hNa$_V$1.8 models contrasting the close overall structural agreement with the extensive outward movement of the VSD$_I$ S4-S5 linker

(dashed black box). **c** (left) Intracellular view of the pore domain showing movements of the VSD$_I$ S4-S5 linker and lower S6$_I$ helix, with (right) close-up views highlighting the angular displacements (indicated by color-coded arrows) of the VSD$_I$ S4-S5 linker and S6 helix. Displacements between Class I and Class II measure ~9 Å for the VSD$_I$ S4-S5 linker and ~4 Å for the S6 helix **d** Model-to-map fits of the VSD$_I$ S4-S5 linker for all apo-hNa$_V$1.8 structures.

ProTx-I is partly buried in the detergent micelle (that mimics the membrane lipid bilayer). This interaction is mediated by a set of aromatic and aliphatic residues (W5, L6, W27, W30) that together form a 'hydrophobic patch', which is commonly observed in ICK peptides[43] (Fig. 4a; Supplementary Movie 3) and explains prior observations that ProTx-I shows some affinity for model membranes, especially anionic membranes[22]. Tryptophan, in particular, is known to preferentially bind to the acyl carbonyl groups at the lipid-water interface[44]; these residues are proposed to anchor the peptide to the membrane and orientate it for interaction with the Na$_V$ VSDs[45]. ProTx-I shows only small conformational changes with respect to its unbound structure, as would be expected for an ICK peptide where the disulfide bonding network maintains rigidity in the peptide core (Supplementary Fig. 8a). The C-terminus is repositioned so as not to clash with the S3-S4 loop and permits the F34 sidechain access to the membrane.

The hNa$_V$1.8-ProTx-I structure shows several points of interaction between the peptide and hNa$_V$1.8 VSD$_II$ (Fig. 4). The membrane-embedded W27 sidechain partially inserts in the cleft formed by the S1/S2 and S3/S4 segments adjacent to I702 on S2 and G745 on S3, while the V29 sidechain is positioned directly on top of the S3 helix at V746/A747, likely hindering movement of this segment during activation (Fig. 4b). The structure places the S3$_II$ V746 sidechain directly below ProTx-I, and its replacement by the bulkier leucine in hNa$_V$1.1-1.7 may

hinder peptide binding and contribute to the slight increase in potency against Na$_V$1.8.

The VSD$_II$ S3-S4 loop in the apo-hNa$_V$1.8 map is of relatively lower resolution, which made tracing the loop backbone challenging; by contrast, the local resolution in this region of the hNa$_V$1.8-ProTx-I complex map was improved (likely due to stabilization from the interaction with ProTx-I) and allowed straightforward tracing of the S3-S4 loop backbone at a higher threshold level (Supplementary Fig. 8b). Comparing the two structures shows that the binding of the toxin induces an inward movement of the top of the S3 helix together with a corresponding movement of the S3-S4 linker towards the pore domain (Fig. 4c and Supplementary Movie 4). This redirection of the S3-S4 linker due to ProTx-I binding propagates along its length such that G750 and S751 now sit directly atop S4 and adjacent to the pore domain, potentially hindering translocation of S4. This movement positions two adjacent lysine sidechains on the S3-S4 linker (K748 and K749, which are unique to hNa$_V$1.8; see Discussion) upwards; the first of these lysine sidechains is positioned close to polar residues on ProTx-I (D31 and S22) where it may form hydrogen-bonding interactions. These observations are consistent with structure-activity studies of ProTx-I. A tethered-toxin alanine scan of ProTx-I against Na$_V$1.7 identified multiple residues that significantly modified peptide activity (including L6, W27, V29, W30, and D31)[26]. The structure shows that many of these residues either form direct contacts with the channel or

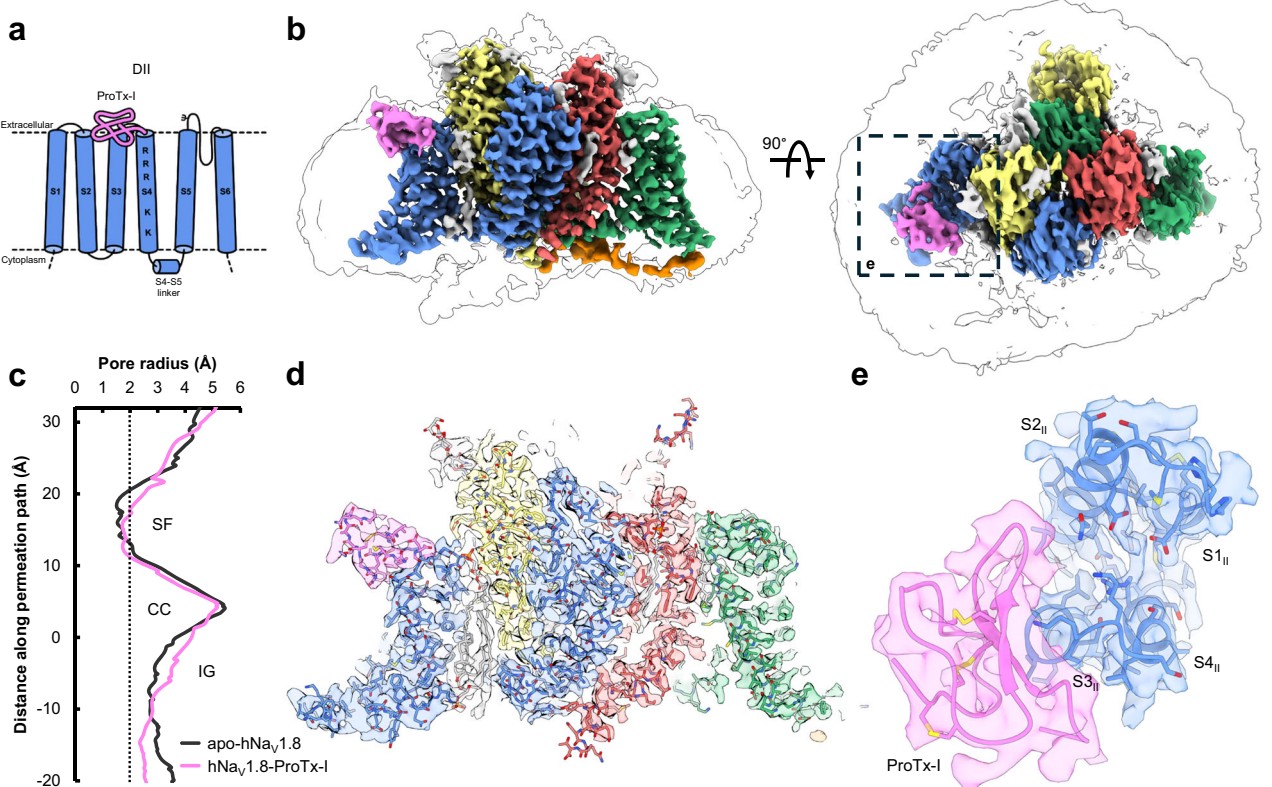

**Fig. 3 | Overall architecture and reconstruction of hNaᵥ1.8-ProTx-I complex.**
**a** Schematic showing the positioning of ProTx-I (pink) on hNaᵥ1.8 VSD$_{II}$. **b** Side and extracellular views of the final hNaᵥ1.8-ProTx-I complex map, colored as in Fig. 1c with ProTx-I in pink; the dashed box highlights the region binding ProTx-I. **c** Pore radius for the hNaᵥ1.8-ProTx-I complex (pink) together with apo-hNaᵥ1.8 (black). **d** Model-to-map fit for a central cross-section of hNaᵥ1.8-ProTx-I, and **e** extracellular view of the fitted map in the ProTx-I-binding region.

form part of the hydrophobic face that anchors the peptide to the membrane (Fig. 4a). Intriguingly, performing the same experiment with Naᵥ1.2 shows an expanded pharmacophore compared with Naᵥ1.7, with R3, W5, S22, R23, G32 also contributing to channel inhibition and suggesting that ProTx-I can adopt different binding modes depending on the Naᵥ isoform that it is targeting[24].

Mutagenesis experiments focusing on the channel have also explored the Naᵥ-ProTx-I interaction. An alanine scan of S3-S4 in a Naᵥ1.2-VSD$_{II}$/Kᵥ1.2 chimera revealed several residues that modulated ProTx-I inhibition[46]. Mapping these residues to the hNaᵥ1.8-ProTx-I structure provides a partial justification of these results. Significant reductions in potency were observed on mutation of the hydrophobic residues at the top of S3$_{II}$; these correspond to V746 and A747 in Naᵥ1.8, and which are directly involved in ProTx-I binding (Fig. 4b). Large changes were also observed for residues at the top of S4$_{II}$, equivalent to S751 and S753 in hNaᵥ1.8, which do not directly interact with ProTx-I in our structure but may instead relate to the inward push on the S3-S4 linker after binding.

Taken together, these structures justify prior structure-activity data as well as the observed pharmacological properties of ProTx-I on voltage-gated sodium channels. ProTx-I is observed to wrap around the top of the S3$_{II}$ helix, and inhibition is also likely mediated by the inward shift of the VSD$_{II}$ S3-S4 linker, which potentially restricts movement of the S4$_{II}$ helix. The positioning of residues which, in the hNaᵥ1.8-ProTx-I structure, do not form direct interactions, also provides hints as to the relative promiscuity of ProTx-I towards hNaᵥ isoforms (discussed below).

## Discussion

This study reports the cryoEM structures of apo-hNaᵥ1.8 and a hNaᵥ1.8-ProTx-I complex and provides insights into the mechanism of channel inhibition by ProTx-I, as well as a useful point of comparison with other structures. ProTx-I was observed to bind to VSD$_{II}$, with no density observed around VSD$_{IV}$, despite evidence from mutagenesis experiments[26]. The addition of ProTx-I seemed to stabilize hNaᵥ1.8 resulting in a better quality and higher resolution map, including in the peptide-binding region of the channel.

The decision to co-express with β-subunits was motivated by low expression yields of hNaᵥ1.8 and poor particle distribution on the grid; we were also keen to minimize the volume of cell culture required to obtain sufficient particles for cryoEM reconstruction, which can reach 40 L in some cases[47]. The β4-subunit was selected based on evidence that it interacts with hNaᵥ1.8 and affects activation and inactivation potentials[35]. However, no significant increase in the expression yield was seen and in subsequent purification and data collection steps only the hNaᵥ1.8 α-subunit was identified. A cryoEM structure of hNaᵥ1.1 together with β4 was able to resolve the β4 extracellular domain even as the co-expressed β3-subunit was not visible and demonstrated direct linking of β4 to the Naᵥ1.1 α-subunit via a disulfide bond to the VSD$_{II}$ S5-S6 extracellular loop[39]. hNaᵥ1.8 lacks the counterpart cysteine at this position required for disulfide bonding, which is likely to significantly weaken the interaction with β4; further investigation will be required to reveal the mechanism of gating modification of hNaᵥ1.8 by β4.

Initial preparations produced homogeneous and good-quality particles, but in insufficient amounts to proceed with cryoEM. Since biochemical approaches did not significantly improve the yield of hNaᵥ1.8, and we wished to avoid more drastic interventions (such as chimeras), we investigated different types of grids to optimize particle density and quality. The use of support films can drastically increase particle retention after blotting compared with conventional holey carbon grids[48] and we found that grids with monolayer graphene

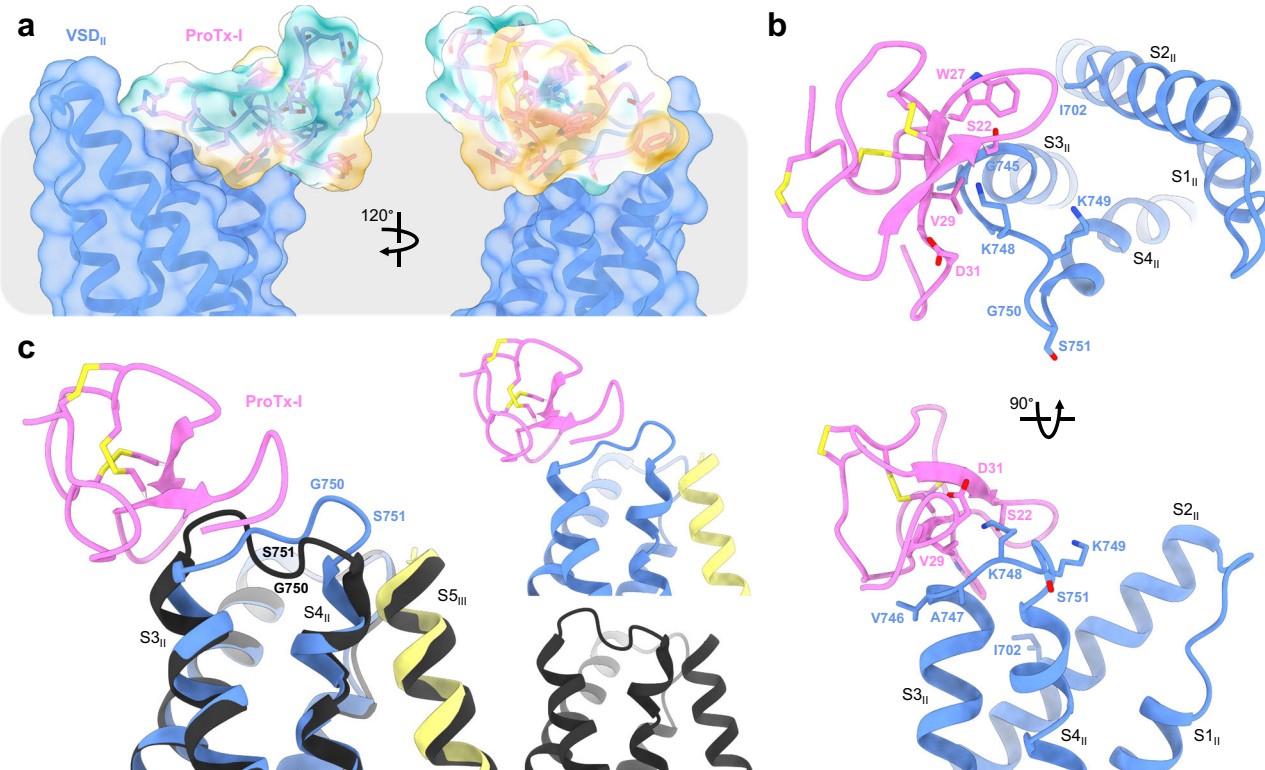

**Fig. 4 | Interactions of ProTx-I with hNa$_V$1.8 and the membrane. a** ProTx-I surface colored by hydrophobicity showing insertion of the hydrophobic patch into the membrane region (gray). Polar residues are indicated in teal, hydrophobic residues are indicated in gold. **b** Two views of the VSD$_{II}$-ProTx-I binding interface showing W27 sidechain partially inserting into the S2/S3 cleft, V29 sitting atop S3$_{II}$ at V746 and A747, and the K748 sidechain on the VSD$_{II}$ S3-S4 linker in range to interact with the ProTx-I D31 and S22 sidechains. **c** (left) Overlay of the VSD$_{II}$ S3-S4 linker position in the hNa$_V$1.8-ProTx-I complex (color scheme as in Fig. 3a) and apo-hNa$_V$1.8 (black) showing inward movement towards the S4 helix, and (right) separated comparison of S3 and S4 helix positions.

support showed the most initial promise. Further optimization of the freezing conditions led to grids with a homogenous distribution of particles (Supplementary Fig. 3c) obtained from as low as 1.5 L cell culture and ultimately resulted in all the high-resolution reconstructions presented in this paper.

The structures of apo-hNa$_V$1.8 revealed a large hinging movement of the VSD$_I$ S4-S5 linker that was resolvable in two separate classes (Fig. 2). This VSD$_I$ S4-S5 linker movement is not observed between the previously determined apo-hNa$_V$1.8 and hNa$_V$1.8-A-803467 complex structures[36], despite the flexibility shown in VSD$_I$ S1-S4. By contrast the VSD$_I$ S4-S5 linker is consistently found tucked in closely to VSD$_{II}$ S5 and S6, as it is in other Na$_V$ structures, while the S6$_I$ helix is also positioned more closely into the pore. Notably, however, similar position shifts are observed between wild-type hNa$_V$1.7 and hNa$_V$1.7-M11[42]. Previous work ascribed the comparative flexibility of VSD$_I$ to unique mutations in hNa$_V$1.8 VSD$_{II}$ S5 and identified two mutants (K806M and L809F) near the VSD$_{II}$ S5-VSD$_I$ interface that individually and collectively shift the voltage of activation to more polarized potentials[36]. Since we were unable to resolve VSD$_I$ in our structures we are unable to confirm the influence of these residues on VSD$_I$ flexibility, but we do observe an inward and upward shift of the VSD$_{II}$ S5 and S6 helices that justifies the connection between VSD$_I$ positioning and channel gating properties.

The unusual movements of the VSD$_I$ S4-S5 linker and S6 helix seen in our structures prompted closer examination of this region. Both regions are highly conserved across human Na$_V$1.1-1.8, although hNa$_V$1.9 shows lower sequence identity (Supplementary Fig. 1). Aside from hNa$_V$1.9, only hNa$_V$1.8 has non-conserved residues in the VSD$_I$ S4-S5 linker, with Val instead of Thr at position 234, and His replacing Glu at position 241. V234 points away from the rest of the channel and does not form any interactions except to solvent or detergent, but H241 is

orientated towards the conserved E402 and Q403 residues on VSD$_I$ S6 (Supplementary Fig. 9). When the VSD$_I$ S4-S5 linker is in the conventional tucked position, the His/Glu sidechain is close enough to interact with these polar residues[36]. Both Glu and His are capable of simultaneously donating and accepting hydrogen bonds, but the imidazole ring on the His sidechain imposes additional geometric restraints; the Glu-His mutation observed at this position in hNa$_V$1.8 may therefore affect the ability to form stabilizing interactions with VSD$_I$ S6 and may contribute to the lability of this region that was observed in our data.

The hNa$_V$1.8-ProTx-I structure is obtained at higher resolution than apo-hNa$_V$1.8, which allows an additional extracellular loop to be modeled into this map. The complex structure demonstrates binding of the peptide to the channel by wrapping around the top of the S3$_{II}$ helix. This interaction is mediated by anchoring of the peptide to the membrane via an external hydrophobic patch, together with acidic and polar residues that can potentially form hydrogen bonds with a lysine residue (unique to Na$_V$1.8) on the VSD$_{II}$ S3-S4 linker. The binding of ProTx-I induces an inward shift of the VSD$_{II}$ S3-S4 linker such that it partially repositions on top of the S4$_{II}$ helix, which we hypothesize hinders the movement of S4$_{II}$ during activation and justifies the gating-modifier properties of ProTx-I.

Comparing the hNa$_V$1.8-ProTx-I structure to other structures of gating-modifier peptides bound to Na$_V$s shows some similarities and differences in their modes of action. A cryoEM study of the gating-modifier peptide Huwentoxin-IV in complex with a nanodisc-bound NaChBac-Na$_V$1.7-VSD$_{II}$ chimera shows that the peptide is similarly orientated by its membrane-inserted hydrophobic patch to present polar residues towards the channel, particularly the K32 sidechain 'stinger' which is proposed to enter the VSD$_{II}$ cleft and come into

proximity with negatively charged residues E822, D827, and E829[49]. This stinger interaction mechanism is maintained when Huwentoxin-IV binds to the channel in the resting conformation[32]. Notably, in hNa$_V$1.8, D827 is modified to lysine while E829 is replaced by glycine; hNa$_V$1.8 additionally has a second lysine at K748, replacing valine in hNa$_V$1.7. The replacement of so many negatively charged residues in hNa$_V$1.7 by positive or neutral residues in Na$_V$1.8 would be sufficient to abolish these interactions and explain why hNa$_V$1.8 is resistant to Huwentoxin-IV[50]. It also justifies the lack of a similar 'stinger' strategy by ProTx-I in its inhibition of hNa$_V$1.8.

A similar chimeric strategy was used to obtain structures of *Thrixopelma pruriens* ICK peptide Protoxin-II in complex with Na$_V$1.7-VSD$_{II}$, which additionally revealed both activated and deactivated conformations[31]. While ProTx-I shows only mild selectivity towards Na$_V$1.8 compared with other isoforms, Protoxin-II is notable for its potency (IC$_{50}$ = 0.3 nM) and selectivity (>100-fold) in favor of hNa$_V$1.7. As with the hNa$_V$1.8-ProTx-I structure, a prominent tryptophan side-chain partitions into the VSD$_{II}$ S2-S3 cleft where it interacts with nearby hydrophobic residues (Fig. 4b). Similar to Huwentoxin-IV, Protoxin-II inserts a lysine residue sidechain to interact with E811, but additionally projects its R22 sidechain towards the acidic residues on the S3-S4 linker. The involvement of the arginine sidechain is of interest because ProTx-I also has an arginine at an equivalent position (R23), and which was identified as an important residue in targeting Na$_V$1.2[24]. The map density for this loop is not sufficient for confident placement of the arginine sidechain (suggestive of regional flexibility), but the hydrogen-bonding partner residues E694 and Q698 on S2$_{II}$ are within range for these interactions to occur and present an additional possible binding mode for Na$_V$ subtypes with acidic and polar residues in these positions. Notably, in hNa$_V$1.7 the equivalent positions (E694 and Q698) are replaced by Lys and Ala, respectively, which would abolish any potential interactions with ProTx-I R23.

Here, the cryoEM reconstructions of apo- and ProTx-I-bound hNa$_V$1.8 provide important insights into the versatile mechanisms that ProTx-I and other gating-modifier peptides have at their disposal to affect Na$_V$ gating. This was enabled by the use of monolayer graphene support that allowed 3D reconstructions from relatively low concentrations of protein stemming from small volumes of cell culture, which may be beneficial for studies of other hNa$_V$s and similarly challenging proteins. These results will assist in the development of analgesic drugs targeting hNa$_V$1.8, as well as aiding the structural characterization of peptide-Na$_V$ complexes yet to be determined.

## Methods

### Isoform and cloning of hNa$_V$1.8 and β4
The hNa$_V$1.8 sequence (Supplementary Fig. 1) used in this study was obtained from GenBank (NM_006514.3; the current canonical sequence NM_006514.4 has M1713 in place of Val, which does not significantly affect the electrophysiological properties of the channel)[36]. The hNa$_V$1.8 sequence was N-terminally tagged with FLAG-tag, Twin-Strep-tag and a TEV protease site. The β4 sequence used in this study was obtained from GenBank (NM_174934.3) and C-terminally tagged with a TEV protease site and 6xHis-tag. The codon-optimized DNA was cloned into pcDNA3.1(+) and purchased from GenScript (genscript.com).

### Transient expression of hNa$_V$1.8 and β4
HEK293 cells (FreeStyle 293-F, Gibco) were seeded at ~0.3 × 10$^6$ cells/mL into 3 L of FreeStyle 293 Expression Medium (Gibco) in a baffled polycarbonate 5 L Erlenmeyer flask and incubated at 37 °C with 8% CO$_2$ at 110 rpm. After 3 days, fresh media prewarmed to 37 °C was added to dilute the cells to 2 × 10$^6$ cells/mL. A total of 1.1 mg/L of DNA was used at a 2:1 ratio of hNa$_V$1.8 to β4 and was mixed into 90 mL of Opti-MEM Reduced Serum Medium (Gibco). A total of 10 mL (1 mg/mL in PBS) of PEI Max 40 kDa (Polysciences Inc.) was added to the DNA and incubated

for 20 min at room temperature. The cells were transiently co-transfected and harvested after 42 h at 800 × g for 30 min at 4 °C. The ~35–40 g wet cell pellet was flash frozen in liquid nitrogen and stored at −80 °C.

### Protein purification of apo-hNa$_V$1.8
A 35 g HEK293 cell pellet, equivalent to 3 L of cells, was homogenized in 60 mL of buffer A (165 mM NaCl, 27.5 mM HEPES pH 7.5, 2 mM MgCl$_2$, 11% glycerol, 10 mM EDTA, and 3 x Pierce Protease Inhibitor Tablets (Thermo Scientific) supplemented with 5 units/mL of Benzonase (Millipore)) by plunging on ice in a glass Dounce tissue grinder with a large clearance pestle. The homogenate was diluted to 120 mL with buffer A and plunged on ice again with a small clearance pestle. The homogenate was diluted further with buffer A to a protein concentration of ~11 mg/mL (as determined by spectrophotometry using a Thermo Scientific NanoDrop). The membrane was solubilized for 2 h at 4 °C on a roller shaker at 30 rpm in 1% n-dodecyl-B-D-maltoside (DDM) (Gold-Bio), 0.2% CHS (Anatrace) by adding 10X solubilization buffer (10% DDM, 2% CHS) for a final protein concentration of ~10 mg/mL. The bulk of the cellular debris was pelleted at 4347 × g using an Eppendorf 5910 R centrifuge for 10 min at 4 °C. The supernatant was clarified further by ultracentrifugation with a Beckman Optima L ultracentrifuge equipped with a SW 32 Ti rotor at 25,000 rpm (r$_{av}$ 76,800 × g) for 30 min at 4 °C.

A 2 mL column volume (CV) of ANTI-FLAG M2 Affinity Gel (Millipore) was equilibrated in a gravity flow column with 2 CVs of buffer B (150 mM NaCl, 25 mM HEPES pH 7.5, 0.06% (w/v) glyco-diosgenin (GDN) (Anatrace)). The supernatant was mixed with the M2 affinity gel for 1 h at 4 °C on a roller shaker at 5 rpm. After collecting the flow through by gravity, the affinity gel was washed gradually into buffer B in the following 5 CVs buffer A and buffer B ratios: 50:50, 25:75, 12.5:87.5, 5:95 and 0:100. Protein was eluted by mixing the M2 affinity gel with 5 CVs of buffer B supplemented with 200 μg/mL of FLAG peptide for 30 min at 4 °C.

For size exclusion chromatography, two eluate fractions were loaded onto a Superose 6 Increase 10/300 GL column (Cytiva) connected to an ÄKTA pure system (Cytiva) in buffer C (150 mM NaCl, 25 mM HEPES pH 7.5, 0.006% (w/v) GDN) (Supplementary Fig. 2a). Eluate 1 consisted of the first 2 mL eluted from the FLAG column; Eluate 2 consisted of the remaining FLAG eluate, concentrated to 0.5 mL in a 4 mL 100 kDa MWCO Amicon Ultra centrifugal filter (3000 × g at 4 °C). The flow rate was 0.7 mL/min at 4 °C. Fractions 12-17 (11.8–14.8 mL) from both eluates were pooled and concentrated as before. The pooled fractions were again purified by size exclusion chromatography using the above method. Finally, fractions 13–15 (12.4–13.9 mL) were pooled and concentrated in a 0.5 mL 100 kDa MWCO Amicron Ultra centrifugal filter (3000 × g at 4 °C) to 60 μL at ~0.4 mg/mL.

### Addition of ProTx-I to apo-hNa$_V$1.8
Performed similarly to the forementioned protocol with the following changes. A 20 g HEK293 cell pellet, equivalent to 1.5 L of cell culture, was homogenized. Fractions 13–15 (12.4–13.9 mL) were pooled and concentrated to 250 μL at ~0.07 mg/mL. The concentrated apo-hNa$_V$1.8 was mixed with 25 μL of 0.5 mM (2 mg/mL) ProTx-I (Smartox Biotechnology) in 1 M HEPES pH 7.4 for a final concentration of 45 μM and incubated on ice for 30 min. The mixture was concentrated in a 0.5 mL 100 kDa MWCO Amicon Ultra centrifugal filter (3000 × g at 4 °C) to 100 μL at ~0.15 mg/mL.

### Negative staining
All samples were negatively stained following an established protocol[51]. Briefly; 3 μL of sample, ranging between 0.01 and 0.05 mg/mL, was pipetted onto glow-discharged carbon-coated 200-mesh Gilder Cu grids (Ted Pella). Excess sample was removed

with filter paper, washed 5 times with 50 μL Milli-Q water drops, and finally stained with two 50 μL drops of 0.75% uranyl formate (Electron Microscopy Sciences) and excess stain was vacuum aspirated. Grids were carbon-coated using a Leica ACE200, negatively glow charged using a PELCO easiGlow (Ted Pella) prior to addition of sample and stain was freshly prepared. All grids were imaged with a JEOL JEM-2100F TEM equipped with a Gatan OneView 4k × 4k camera. Negative stain 2D class averages (Supplementary Fig. 2d) were calculated using RELION 3.1[52].

### CryoEM grid freezing
Quantifoil R2/4 300 mesh Au grids with monolayer graphene support film (Graphenea) were negatively glow discharged using a PELCO easiGlow (Ted Pella) with the monolayer graphene (front) facing up. A Leica EM GP2 set to 10 °C and 96% humidity was used to freeze the grids. For apo-hNa$_V$1.8, 0.4 mg/mL sample was diluted with buffer C to 0.25 mg/mL and 3 μL was applied to the front of the grid and incubated for 60 s before front blotting for 3 s. For the hNa$_V$1.8-ProTx-I complex, 3 μL of 0.15 mg/mL sample was applied to the front of the grid and blotted as before. Grids were plunge frozen in liquid ethane and stored in liquid nitrogen.

### CryoEM data collection
All movies were collected with a 300 kV FEI Titan Krios microscope equipped with a Gatan K3 direct electron detector. For apo-hNa$_V$1.8, super-resolution movies were collected using SerialEM[53] at a pixel size of 0.839 Å/pixel with a total dose of 60 e$^-$/Å$^2$ spread over 60 total frames, with a defocus range of −1 to −2.5 μm and a 100 μm objective aperture. Energy filter slit width was set to 20 eV. The hNa$_V$1.8-ProTx-I acquisition was performed similarly with the following changes: data was collected at a pixel size of 0.827 Å/pixel with a defocus range of −1 to −2 μm. Full data collection parameters are highlighted in Supplementary Table 1.

### CryoEM data processing of apo-hNa$_V$1.8
The processing pipeline is described in Supplementary Fig. 4. Briefly; 13,124 movies were imported into CryoSPARC 4.2[54] for patch motion correction[55] and patch contrast transfer function (CTF) estimation. 13,006 micrographs were selected for blob picking using circular and elliptical templates resulting in 4,878,994 particle coordinates. Selected 2D classes were utilized for template picking resulting in 11,792,840 particle coordinates. After multiple rounds of 2D classifications, a subset of particles showing different views of apo-hNa$_V$1.8 were selected for ab initio initial 3D model building. Two 3D classes were selected and used as references to parse particles via heterogeneous refinement. Eventually 67,333 particles were used for non-uniform refinement[56] to create a 3.5 Å map. This map was used for template picking resulting in 12,107,700 particle coordinates. Subsequent 2D classifications, 3D refinements, and 3D classifications resulted in a non-uniform refined and sharpened reconstruction at an overall resolution of 3.3 Å from 119,211 particles. This map was used for the final template picking resulting in 12,424,068 particle coordinates.

The 12,424,068 particles resulting from the final template pick were used in subsequent 2D classifications, 3D refinements, and 3D classifications which resulted in a non-uniform refined and sharpened reconstruction at an overall resolution of 3.2 Å from 120,821 particles. 3D variability analysis[57] resulted in maps with varied conformations from distinct particles which were used for ab initio initial 3D model building. Two classes were subjected to non-uniform refinement and resulted in two distinct conformations of the VSD$_I$ S4-S5 linker as Class I with 84,466 particles and Class II with 82,542 particles at overall resolutions of 3.24 Å and 3.22 Å, respectively. The initial 3D model of Class II was used to refine a final sharpened map of apo-hNa$_V$1.8 with all 120,821 particles at an overall resolution of 3.12 Å.

### CryoEM data processing of hNa$_V$1.8-ProTx-I
The processing pipeline is described in Supplementary Fig. 6. Processing of the hNa$_V$1.8-ProTx-I dataset was performed similarly to the apo-hNa$_V$1.8 dataset with the following changes. 15,400 movies were processed using CryoSPARC 4.4. Template picking using the final apo-hNa$_V$1.8 map resulted in 10,509,847 particle coordinates for subsequent processing. After multiple rounds of 2D classifications, a subset of particles showing different views of hNa$_V$1.8-ProTx-I were selected for ab initio initial 3D model building. A subset of 197,157 particles were used for non-uniform refinement into a 3D reconstruction at an overall resolution of 2.9 Å. Subsequent 2D classifications, 3D refinements, and 3D classifications resulted in a non-uniform refined and sharpened hNa$_V$1.8-ProTx-I reconstruction at an overall resolution of 2.76 Å from 267,708 particles. Focus refinement of VSD$_{I-II}$, as well as 3D classification did not aid in resolving VSD$_I$ or increasing the resolution of the ProTx-I binding region.

### Model building, refinement and validation
hNa$_V$1.8 from PDB 7WFW[36] was rigid body fitted into the final apo-hNa$_V$1.8 map using ChimeraX. No density for β4 was observed and therefore was not modeled. The apo-hNa$_V$1.8 model was modified with M1713V, glycosylation sites were adjusted as necessary and the VSD$_I$ S4-S5 linker was positioned to best fit the map using *Coot*[58]. Additionally, 7WFW was found to contain a mutation (S894F) that differs from the canonical hNa$_V$1.8 sequence (NM_006514.3 and NM_006514.4), which was updated in our model. Cholesterol and bound lipids from PDB 7WE4 were used for model building. The model was refined in *Coot* and subsequently refined against the corresponding map using *Phenix* real-space refinement[59]. Models for Class I and II were initially built using an earlier apo-hNa$_V$1.8 model and similarly refined as described.

Modeling for hNa$_V$1.8-ProTx-I used an earlier apo-hNa$_V$1.8 model along with a single ProTx-I model from the NMR ensemble PDB 2M9L[24] and both were rigid body fit into the hNa$_V$1.8-ProTx-I map using ChimeraX[60–62]. Multiple orientations of ProTx-I were sampled to optimize the model-to-map fit. Steps for adjustments and refinements were performed similarly to apo-hNa$_V$1.8.

Model validations were performed using *Phenix* and MolProbity[63,64]. Statistics are available in Supplementary Table 1. Pore path and radius were determined using MOLEonline[65]. All figures were prepared with UCSF ChimeraX, Fiji[66], Adobe Photoshop and Microsoft PowerPoint. Supplementary movies were prepared with UCSF ChimeraX.

**Antibody**. The FLAG Tag antibody, HRP-conjugated, mouse Ab; GenScript cat #A01428-100; lot #21O5K001

### Reporting summary
Further information on research design is available in the Nature Portfolio Reporting Summary linked to this article.

## Data availability
The cryoEM maps associated with this study have been deposited to the Electron Microscopy Data Bank (EMDB) under accession codes EMD-46718 (apo-hNa$_V$1.8), EMD-46719 (Class I), EMD-46720 (Class II), and EMD-46721 (hNa$_V$1.8-ProTx-I). The atomic coordinates associated with this study have been deposited to the Protein Data Bank (PDB) under the accession codes 9DBK (apo-hNa$_V$1.8), 9DBL (Class I), 9DBM (Class II), and 9DBN (hNa$_V$1.8-ProTx-I). Previously reported cryoEM maps and models referred to in this manuscript can be found under accession codes 2M9L (Solution structure of protoxin-1), EMD-32476 (Apo human Nav1.8), 7WFW (Apo human Nav1.8), and 7WE4 (Human Nav1.8 with A-803467, class I). Source data for Figs. 2a, 3c, and Supplementary Fig. 2a-c are provided with this paper. The data that

 

support this study are available from the corresponding authors upon request. Source data are provided with this paper.

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

## Acknowledgements

The authors thank all members of the Gonen lab for helpful and critical discussions. The Gonen lab and this research was supported by the Defense Threat Reduction Agency HDTRA1-21–1-0004 and the National Institute of General Medical Sciences R35-GM142797.

## Author contributions

B.N. performed the experimentation and cryoEM processing. B.N. and S.G. experimental design. B.N. with the guidance of S.M. and S.G. performed model building. All authors analyzed and interpreted the data. Project conception and work supervision by S.G. S.M. drafted an early version of the manuscript. All authors contributed to and approved the final manuscript.

## Competing interests

The authors declare no competing interests.
