## [Transparent Peer Review file · Nature Communications]

Structural basis of inhibition of human Nav1.8 by the tarantula venom peptide Protoxin-I

Corresponding Author: Professor Shane Gonen

Version 0:

Reviewer comments:

Reviewer #1

(Remarks to the Author)

- What are the noteworthy results?

Voltage-gated sodium channels (hNav1.1-1.9) mediate selective Na⁺ inward current in response to membrane depolarization. One of the Nav isoforms, Nav1.8, plays an important role in nociception and chronic pain. Inhibitors of Nav1.8 may have use as potential therapeutics for pain treatments. Peptide inhibitors including tarantula venom peptide Protoxin-I may have better potential for selective inhibition of Nav channel isoforms. Neumann et.al. report the cryo-EM structures of apo-Nav1.8 and Protoxin-I-bound Nav1.8 complex.

- Will the work be of significance to the field and related fields? How does it compare to the established literature?

Even though a prior apo-Nav1.8 structure (PDB 7WFW) has been reported, in the new apo-structure in this manuscript, the VSD1 S4-S5 linker is positioned differently (by up to 17 Å) compared to 7WFW. The use of monolayer graphene grids (0.4 nm thickness) also allowed the authors to determine the Protoxin-I-bound Nav1.8 complex structure using just ~20 g cell pellet from 1.5 L of cell culture, significantly lower than typical Nav preparations. The Protoxin-I binding site will facilitate structure-based drug design of isoform-selective inhibitors of Nav channels.

- Does the work support the conclusions and claims, or is additional evidence needed?

The data are of high quality and support the conclusions and claims.

- Are there any flaws in the data analysis, interpretation and conclusions? Do these prohibit publication or require revision?

Overall, the data analyses are solid. Supplementary Figure 4: Processing flowchart for apo-hNav1.8 using CryoSPARC. In the data processing workflow, there are three rounds of template picking. Why is that? Could that introduce bias into the reconstruction?

- Is the methodology sound? Does the work meet the expected standards in your field?

The methodology is sound.

- Is there enough detail provided in the methods for the work to be reproduced?

Enough detail is provided in the methods. The use of monolayer graphene grids may facilitate cryo-EM structure determination of other challenging targets.

Minor comments.

1. Fig. 3e shows the side chains of multiple residues in S1(II), S2(II), S3(II) and S4(II), without labeling. Is it necessary to show the side chains?
2. Line 375. "Aside from hNav1.9, only hNav1.8 has any mutations to the VSD1 S4-S5 linker, with Val instead of Thr at position 234, and His replacing Glu at position 241." Are these mutations? Or are they non-conserved residues between Nav isoforms?
3. In the new apo-structure in this manuscript, the VSD1 S4-S5 linker is positioned differently (by up to 17 Å) compared to 7WFW. What could be the reason for the difference?

Reviewer #2

(Remarks to the Author)

Reviewer #3

(Remarks to the Author)

In this manuscript, the authors report cryoEM structures of full-length human NaV1.8 with and without the gating modifier toxin ProTx-I. The study provides insight into the toxin-channel interaction and is useful as a comparison to previous models. Note: My expertise is in toxins and toxin-channel interactions rather than cryo-EM—my appraisal of the technical aspects related to the cryo-EM are therefore limited.

In my opinion, the strength of the study is that it serves as independently generated models of apoNaV1.8 and its interaction with a gating-modifier toxin. These have both been generated before (albeit the latter with different toxin-channel combinations) and it could be argued that this study offers little new insight, but I think it is important for the field to see the previous studies independently “replicated”. The experimental methodology and interpretation of the data appear to be of high quality. I appreciated the transparency in reporting aspects of the study that could be viewed as “flaws” e.g. the unresolved VSDI. In short, I think this study will be a useful addition to the literature.

Comments (in no particular order):

In each of the structural models (apo and toxin-bound) the channel is, as expected, in the inactivated state. The differences between apo and toxin-bound channels could be described as subtle. One interpretation of this is that this simply reflects stabilisation, by the toxin, of the channel in the inactivated state. This is eluded to indirectly in the manuscript, but I think it should be stated (or argued against) more directly with reference to previously reported functional data.

I had some questions about the preparation of the toxin-channel complex (lines 214 -217). Did the authors attempt to incubate the toxin with the cells prior to or during purification or only after? Is it certain that the observed low particle contrast was due to excess ProTx-I—could this be elaborated on briefly. I found the following sentence difficult to interpret: “we therefore introduced ProTx-I prior to a final concentration step to remove unbound peptide”. Could this be reworded and elaborated on please i.e. How does the final concentration step remove unbound peptide?

L155-157: “The quality of the map allowed modeling of glycosylation at several positions on the extracellular loops, as well as possible cholesterol, lipids, and detergent molecules bound to the transmembrane region.” I would have appreciated a few more sentences on this, particularly the location and type of glycosylations (as far as can be resolved) and how this compares with previous models.

L253: “ProTx-I is partly buried in the membrane.” Is it correct to refer to this as a membrane? Personally, I picture a membrane as a lipid bilayer, which is not the case here. Perhaps revise the wording.

Figure 4a. The illustration of where the “membrane” sits seems a bit unscientific. What is this based on? I'd prefer to see the actual experimental density here.

Very minor:

L39: “...give each hNav isoform particular roles in sensation.” The way this is worded implies that all Nav isoforms have a role in sensation (and nothing else). Consider revising.

L44: “...higher persistent current”. “increased/greater” rather than higher.

L68: “ProTx-I is isolated from...”. “was” rather than is.

Version 1:

Reviewer comments:

Reviewer #1

(Remarks to the Author)

The authors have addressed my concerns.

Reviewer #2

(Remarks to the Author)

Reviewer #3

(Remarks to the Author)

The authors have addressed my previous comments. I have no additional comments and congratulate the authors on their

excellent study.

Dear Colleagues,

We thank you for your kind and insightful comments. In light of your suggestions, we have made several additions and clarifications that are summarized below. We also added required sections at the end of the manuscript, updated Table 1 (with included lipids for phenix validation) and found small typos. All changes in the manuscript are highlighted in Blue.

We hope that these additions have deepened our analysis, provided greater clarity, and led to an improved manuscript.

We look forward to receiving further feedback.

Sincerely,

Shane Gonen

Reviewer #1 (Remarks to the Author):

- What are the noteworthy results?

Voltage-gated sodium channels (hNav1.1-1.9) mediate selective Na⁺ inward current in response to membrane depolarization. One of the Nav isoforms, Nav1.8, plays an important role in nociception and chronic pain. Inhibitors of Nav1.8 may have use as potential therapeutics for pain treatments. Peptide inhibitors including tarantula venom peptide Protoxin-I may have better potential for selective inhibition of Nav channel isoforms. Neumann et.al. report the cryo-EM structures of apo-Nav1.8 and Protoxin-I-bound Nav1.8 complex.

- Will the work be of significance to the field and related fields? How does it compare to the established literature?

Even though a prior apo-Nav1.8 structure (PDB 7WFW) has been reported, in the new apo-structure in this manuscript, the VSDI S4-S5 linker is positioned differently (by up to 17 Å) compared to 7WFW. The use of monolayer graphene grids (0.4 nm thickness) also allowed the authors to determine the Protoxin-I-bound Nav1.8 complex structure using just ~20 g cell pellet from 1.5 L of cell culture, significantly lower than typical Nav preparations. The Protoxin-I binding site will facilitate structure-based drug design of isoform-selective inhibitors of Nav channels.

- Does the work support the conclusions and claims, or is additional evidence needed?

The data are of high quality and support the conclusions and claims.

- Are there any flaws in the data analysis, interpretation and conclusions? Do these prohibit publication or require revision?

Overall, the data analyses are solid. Supplementary Figure 4: Processing flowchart for apo-hNav1.8 using CryoSPARC. In the data processing workflow, there are three rounds of template picking. Why is that? Could that introduce bias into the reconstruction?

Authors' response: This was our first time using CryoSPARC and the challenge involved in this dataset led us to try different parameters (described below). We also encountered some hard drive issues early on requiring us to repeat some steps. This contrasts with the hNav1.8-ProTx-I dataset where the processing pipeline (see

Supplementary Figure 6) is much more straightforward due to prior experience (and perhaps also the stabilizing effect of protoxin-I, helping alignment).

The first round of template pick used selected 2D classes as templates. The second round used templates created from a non-uniform refinement assuming it would perform better than the first. The third round used templates created from a higher resolved non-uniform refinement in the hopes of recovering additional particles.

In the end though, we were cautious, and the templates used were filtered to low (20Å) resolution, obscuring high-resolution features and resulting in basically every particle being chosen for classifications and further processing, similar to what one would get from a generic “blob” picker.

As an example, the image below shows results from the 3rd template pick and a blob picker using the same parameters (only without templates).

We have added lines 557-563 to the methods to clarify only particles from the third template pick were used in the final reconstructions.

- Is the methodology sound? Does the work meet the expected standards in your field?

The methodology is sound.

- Is there enough detail provided in the methods for the work to be reproduced?

Enough detail is provided in the methods. The use of monolayer graphene grids may facilitate cryo-EM structure determination of other challenging targets.

Minor comments.

1. Fig. 3e shows the side chains of multiple residues in S1(II), S2(II), S3(II) and S4(II), without labeling. Is it necessary to show the side chains?

Authors' response: We thank the reviewer for their close attention to this figure. While it is not usually advantageous to display side chains without labeling, in this particular case we felt that addition of the side chains improved the clarity of the top-down view of the helices and assisted the reader in understanding the fit and resolution of this region to the map. Given this reasoning, please let us know if this depiction is not helpful in this regard and we can redesign the figure.

2. Line 375. “Aside from hNav1.9, only hNav1.8 has any mutations to the VSDI S4-S5 linker, with Val instead of Thr at position 234, and His replacing Glu at position 241.” Are these mutations? Or are they non-conserved residues between Nav isoforms?

Authors' response: Thank you for pointing this out; we have reworded mutations to non-conserved residues in line 381.

3. In the new apo-structure in this manuscript, the VSD1 S4-S5 linker is positioned differently (by up to 17 Å) compared to 7WFW. What could be the reason for the difference?

Authors' response: Our apo-hNav1.8 structures and 7WFW establish that VSD1 is capable of movement. The observed differences in the VSD1 S4-S5 linker are likely due to the final particle subsets that are used for refinement, in which the VSD1 S4-S5 linker is captured in a particular stage of movement but other conformations along the spectrum may also be present. It should be noted that (unlike the gating-modifier peptide used in our study) when a pore-blocking compound was added, the VSD1 S4-S5 linker was only observed in a single position and VSD1 was resolved.

Reviewer #2 (Remarks to the Author):

Authors' response: This sounds like a fantastic initiative, and we thank reviewer #2 for their time and commitment to reviewing our manuscript.

Reviewer #3 (Remarks to the Author):

In this manuscript, the authors report cryoEM structures of full-length human NaV1.8 with and without the gating modifier toxin ProTx-I. The study provides insight into the toxin-channel interaction and is useful as a comparison to previous models. Note: My expertise is in toxins and toxin-channel interactions rather than cryo-EM—my appraisal of the technical aspects related to the cryo-EM are therefore limited.

In my opinion, the strength of the study is that it serves as independently generated models of apoNaV1.8 and its interaction with a gating-modifier toxin. These have both been generated before (albeit the latter with different toxin-channel combinations) and it could be argued that this study offers little new insight, but I think it is important for the field to see the previous studies independently “replicated”. The experimental methodology and interpretation of the data appear to be of high quality. I appreciated the transparency in reporting aspects of the study that could be viewed as “flaws” e.g. the unresolved VSDI. In short, I think this study will be a useful addition to the literature.

Comments (in no particular order):

In each of the structural models (apo and toxin-bound) the channel is, as expected, in the inactivated state. The differences between apo and toxin-bound channels could be described as subtle. One interpretation of this is that this simply reflects stabilisation, by the toxin, of the channel in the inactivated state. This is eluded to indirectly in the manuscript, but I think it should be stated (or argued against) more directly with reference to previously reported functional data.

Authors' response: The reviewer raises an important point. Our use of the word *stabilization* in this context refers to effects of the ProTx-I binding that assist in cryoEM reconstruction; these include reducing local movements to enable refinement to high resolution with improved discernment of map features (such as the ECL₁ loop), and producing a more self-consistent particle dataset that allows more particles to be used in the reconstruction.

These effects are distinct from stabilization of a particular channel conformation (sometimes described as ‘trapping’ a channel in a particular state). In this case the global structural differences between the apo- and peptide-bound conformations are indeed small, because the conformation that the channel adopts in cryoEM experiments is determined by the absence of membrane polarization, and any trapping effects that ProTx-I may exert are insufficient to overcome this effect. It should be noted, however, that we do observe conformational changes in the immediate region of the toxin binding site.

I had some questions about the preparation of the toxin-channel complex (lines 214-217). Did the authors attempt to incubate the toxin with the cells prior to or during purification or only after? Is it certain that the observed low particle contrast was due to excess ProTx-I—could this be elaborated on briefly. I found the following sentence difficult to interpret: “we therefore introduced ProTx-I prior to a final concentration step to remove unbound peptide”. Could this be reworded and elaborated on please i.e. How does the final concentration step remove unbound peptide?

Author's response: We are happy to explain our methodology in further detail and have amended the manuscript accordingly.

Regarding the reviewer's first question, we did not attempt to incubate the cells with ProTx-I prior to beginning the purification, for several reasons. Our purification scheme was inspired by other published protocols that have successfully determined the structures of hNa_v complexes, in which ligands are added during the final step; additionally, binding of the peptide to the channel is promoted by adding high concentrations of peptide to high concentrations of purified hNa_v protein, which is only possible at the later stages of purification. This workflow also allows for other ligands to be screened for binding without restructuring the entire protein purification workflow, which we hope will aid future avenues for our hNa_v program. Outside of biochemical justifications, health and safety considerations require us to minimize the quantities of toxic peptides used, and to limit the number of steps in which toxins are present. Nonetheless, we find the reviewer's suggestion an intriguing avenue

for future research as adding toxins directly to cells may promote binding by beta-subunits, which were not observed after purification.

We observed low particle contrast due to background noise only in samples to which excess ProTx-I was added immediately prior to grid freezing (see Supplementary Figure 3), so we understood that we needed to remove this excess peptide before sample preparation. A simple option is to use the final protein concentration step (using a 100 kDa MWCO membrane) to simultaneously concentrate the hNav1.8-ProTx-I and remove unbound peptide. This method allows us to avoid further size-exclusion chromatography steps that may result in sample loss. We have clarified our approach in lines 215-219 of the manuscript and provided additional details in the Methods section.

L155-157: “The quality of the map allowed modeling of glycosylation at several positions on the extracellular loops, as well as possible cholesterol, lipids, and detergent molecules bound to the transmembrane region.” I would have appreciated a few more sentences on this, particularly the location and type of glycosylations (as far as can be resolved) and how this compares with previous models.

Authors’ response: We have added additional details on the glycosylation sites discernable in our map to lines 155-157. Our maps resolve five N-linked glycosylation sites (residues 312, 819, 1312, 1328, and 1336) together with low-resolution density for the glycans themselves. These details are also visible in previous models, however, unlike previous work, our maps can discern the extracellular loop ECL₁. This loop contains a further two N-linked glycosylation sites (residues 284 and 288), but the resolution was insufficient to model the glycans themselves. We have added new sentences at lines 223-225 and 227-229 to incorporate these details into the manuscript.

L253: “ProTx-I is partly buried in the membrane.” Is it correct to refer to this as a membrane? Personally, I picture a membrane as a lipid bilayer, which is not the case here. Perhaps revise the wording.

Authors’ response: This is a great point. We have reworded line 258 to clarify that, in this experiment, the channel is embedded in a detergent micelle.

Figure 4a. The illustration of where the “membrane” sits seems a bit unscientific. What is this based on? I’d prefer to see the actual experimental density here.

Authors’ response: We appreciate the reviewer’s careful attention to this figure. The experimental density for the detergent micelle in the hNav1.8-ProTx-I complex is shown in Figure 3b. The depiction of the membrane in Figure 4a is simplified so as not to obscure details of the ProTx-I and VSDII surfaces, similar to other structural manuscripts. We have repositioned the membrane representation in Figure 4a to be in closest accordance with our experimental map of the detergent micelle.

Very minor:

L39: “...give each hNav isoform particular roles in sensation.” The way this is worded implies that all Nav isoforms have a role in sensation (and nothing else). Consider revising.

Authors’ response: We thank the reviewer for pointing out the potential for misinterpretation of this phrase. We have revised the manuscript to better reflect the roles of hNav_v isoforms.

L44: “...higher persistent current”. “increased/greater” rather than higher.

Authors’ response: We have revised the manuscript accordingly.

L68: “ProTx-I is isolated from...”. “was” rather than is.

Authors’ response: We have revised the manuscript accordingly.